# Error Correcting Output Codes Improve Probability Estimation and Adversarial Robustness of Deep Neural Networks

**Gunjan Verma**
CCDC Army Research Laboratory
Adelphi, MD 20783
gunjan.verma.civ@mail.mil

**Ananthram Swami**
CCDC Army Research Laboratory
Adelphi, MD 20783
ananthram.swami.civ@mail.mil

## Abstract

Modern machine learning systems are susceptible to adversarial examples; inputs which clearly preserve the characteristic semantics of a given class, but whose classification is (usually confidently) incorrect. Existing approaches to adversarial defense generally rely on modifying the *input*, e.g. quantization, or the learned *model* parameters, e.g. via adversarial training. However, recent research has shown that most such approaches succumb to adversarial examples when different norms or more sophisticated adaptive attacks are considered. In this paper, we propose a fundamentally different approach which instead changes the way the *output* is represented and decoded. This simple approach achieves state-of-the-art robustness to adversarial examples for $L_2$ and $L_\infty$ based adversarial perturbations on MNIST and CIFAR10. In addition, even under strong white-box attacks, we find that our model often assigns adversarial examples a low probability; those with high probability are often *interpretable*, i.e. perturbed towards the perceptual boundary between the original and adversarial class. Our approach has several advantages: it yields more meaningful probability estimates, is extremely fast during training and testing, requires essentially no architectural changes to existing discriminative learning pipelines, is wholly complementary to other defense approaches including adversarial training, and does not sacrifice benign test set performance.

## 1 Introduction

Deep neural networks (DNNs) achieve state-of-the-art performance on image classification, speech recognition, and game-playing, among many other applications. However, they are also vulnerable to *adversarial* examples, inputs with carefully chosen perturbations that are misclassified despite containing no semantic changes [1]. Often, these perturbations are "small" in some sense, e.g. some $L_p$ norm. From a scientific perspective, the existence of adversarial examples demonstrates that machine learning models that achieve superhuman performance on benign, "naturally occurring" data sets in fact possess potentially dangerous failure modes. The existence of these failure modes threatens the reliable deployment of machine learning in automation of tasks [2]. A myriad of defenses have been proposed to make DNNs more robust to adversarial examples; virtually all have been shown to have serious limitations, however, and at present a solution remains elusive [3].

Adversarial defenses that have been proposed to date can broadly be taxonomized by which part of the learning pipeline they aim to protect. *Input*-based defenses seek to explicitly modify the input directly. These are comprised of three main classes of methods: i) manifold-based, which projects the input into a different space ([4] in which the adversarial perturbation is presumably mitgiated, ii) quantization-based, which alters input data resolution [5] or encoding. [6], and iii) randomization-based, in which portions of the input and/or hidden layer activations are randomized

or zeroed out. [7]. *Model*-based defenses seek to alter the learned model by the use of alternative training or modeling strategies. These comprise two main classes of methods: i) adversarial training (augments training data with adversarial examples ([1], [8]), and ii) generative models (closely related to the manifold-based approaches), which seek to model properties of the input or hidden layers, such as the distribution of activations, and detect adversarial examples as those with small probability or activation value [9, 10] under the natural data distribution. *Certification*-based methods aim to provide provable robustness guarantees against the existence of adversarial examples [11, 12].

However, all these approaches have been shown to have serious limitations. Certification-based methods offer guarantees which are either restricted to training examples or yield vacuous guarantees for all but very small $L_p$ perturbation magnitudes. For example, [13] provide certificates for $L_2$ perturbations up to $0.5$; changing a *single* pixel from all black to all white would fall outside this threat model. Input- and model-based defenses are generally effective against white-box attacks (attacker has full knowledge of model) or black-box attacks, but not both [14]. Virtually all approaches successful against white-box attacks mask the gradient of the loss with respect to the input but do not truly increase model robustness [15] Another challenge is that existing defenses are designed against particular attack models (e.g., the attacker will mount a bounded $L_\infty$ attack); these defenses usually fail completely when the attacker model changes (e.g., rotating or translating the inputs [16]).

In this paper, we draw inspiration from coding theory, the branch of information theory which studies the design of codes to ensure reliable delivery of a digital signal over a noisy channel; here, we draw an analogy between the signal and the output (label) encoding, and between the noisy channel and adversarial perturbation [1]. At a high level, coding theory formalizes the idea that in order to minimize the probability of signal error, codewords should be "well-separated" from one another, i.e., differ in many bits. In this paper, we find that encoding the outputs using such codes, as opposed to the conventional one-hot encoding, has some surprising effects, including significantly improved probability estimates and increased robustness to adversarial as well as random or "fooling" examples [17] (e.g., noise-like examples classified with high confidence as belonging to some class). The main contributions of our paper are threefold:

- We demonstrate why standard one-hot encoding is susceptible to adversarial and fooling examples and prone to overconfident probability estimates.

- We demonstrate that well-chosen error-correcting output codes, coupled with a modified decoding strategy, leads to an intrinsically more robust system that also yields better probabillity estimates.

- We perform extensive experimental evaluations of our method on L$_2$ and L$_\infty$ based adversarial perturbations which show our approach achieves or surpasses state-of-the-art results.

In contrast to existing work, our method is *not explicitly designed* to be a defense against adversarial examples or to generate meaningful probability estimates. Rather, we find that these phenomena are emergent properties of properly encoding the class labels. Unlike existing approaches, ours is extremely easy to implement, requires far fewer model parameters, is fast to train and to execute during inference time, and is completely complementary to existing defenses based on adversarial training [8] or generative models [18].

## 2   Model framework

We first define some notation. We denote by $\mathbf{C}$ the $M \times N$ matrix of codewords, where $M$ denotes the number of classes and $N$ the codeword length. The $k$th row of $\mathbf{C}$, $\mathbf{C}_k$, is the desired output of the DNN when the input is from class $k$. For typical one-hot encoding, $\mathbf{C} = \mathbf{I}_M$, the identity matrix of order $M$. Other choices of $\mathbf{C}$ are often denoted by the general term "error-correcting output code" (ECOC) and have been studied mainly in the context of improving a learner's (non-adversarial) generalization performance [19]. In this paper, we will consider codes with $N = M$ and $N > M$. Also. we define a "sigmoid" function as a general "S-shaped" monotonically non-decreasing activation function which maps a scalar in the reals $\mathbb{R}$ to some fixed range such as $[0, 1]$ or $[-1, 1]$: In this paper, we will find use for two sigmoid functions: the "logistic" function, defined in Section 2.2, and the "tanh", the hyperbolic tangent function.

## 2.1 Softmax Activation

The choice $\mathbf{C} = \mathbf{I}_M$ along with the softmax activation are two nearly universally adopted components for multi-class classification. The softmax maps a vector $\mathbf{z}$ in $\mathbb{R}^M$ tnto the $(M-1)$-dimensional probability simplex. We will denote the $M$-dimensional vector of softmax activations by $\psi$. The $k$th softmax activation is given by

$$p_\psi(k) = \frac{\exp(z_k)}{\sum_{i=1}^{M} \exp(z_i)} \tag{1}$$

$\mathbf{z}$ is often referred to as the vector of logits. Figure 1(a) plots $p_\psi(0)$ as a function of $\mathbf{z}$ for the case $M = 2$ of two classes. Class 0 has one-hot codeword $(1,0)$ and class 1 has codeword $(0,1)$. The $x$ axis denotes the logit $z_0$ and the $y$ axis denotes the logit $z_1$. The amount of red is proportional to $p_\psi(0)$; i.e., dark red indicates $p_\psi(0) \approx 1$ and dark blue indicates $p_\psi(0) \approx 0$. Unsurprisingly, the figure shows that the softmax assigns highest probability to the class whose corresponding logit is largest. Importantly, the softmax is able to express uncertainty between the two classes (i.e., assign roughly equal probability to both classes) only along the diagonal, i.e. when $z_0 \approx z_1$. In higher dimensional spaces, where $M > 2$, the softmax is uncertain between any two classes $i$ and $j$ (i.e. $p_\psi(i) \approx p_\psi(j)$) if and only if the corresponding logits $z_i$ and $z_j$ are approximately equal. The region $z_i \approx z_j$ is "almost" a hyperplane, a $M-1$ dimensional subspace of $\mathbb{R}^M$ which has negligble volume. Thus, from the perspective of representing uncertainty, the softmax suffers from a fatal flaw: it is certain almost everywhere in logit space. For very accurate models applied to non-adversarial inputs (the classical setting considered by machine learning), this is acceptable since the model will typically be correct and confident. But on adversarial inputs, for which the model is incorrect, it will often still be confident; indeed it is this (over) confidence that is the central challenge posed by adversarial examples. We will see further evidence of this phenomena in Section 3.

## 2.2 Sigmoid Activation

We now propose an alternative way to map logits to class probabilities. The essential idea is simple: the model maps logits to the elements of a codeword and assigns probability to class $k$ as proportional to how positvely correlated the model output is to $\mathbf{C}_k$.

$$p_\sigma(k) = \frac{\max(\sigma(\mathbf{z}) \cdot \mathbf{C}_k, 0)}{\sum_{i=1}^{M}(\max(\sigma(\mathbf{z}) \cdot \mathbf{C}_i, 0))} \tag{2}$$

Here, $\sigma(\mathbf{z})$ and $\mathbf{C}_k$ are length-N vectors. Here, $\sigma$ is some sigmoid function which is applied element-wise to the logits. For example, the logistic function has $k$th output as $\sigma_k(\mathbf{z}) = \frac{1}{1+\exp(-z_k)}$ taking values in $(0,1)$. Another possible choice for $\sigma$ is the tanh function taking values in $(-1,1)$. When $\mathbf{C}$ take values in $\{0,1\}$, then the logistic function is appropriate to use; in this case, the max operation is unnecessary. However if $\mathbf{C}$ take values in $\{-1,1\}$ then the tanh function is used and the max operator is needed to avoid negative probabilities. Equation (2) is intuitive; it computes the probability of a class as proportional to how similar (correlated) the model's predicted code $\sigma(\mathbf{z})$ is to each codeword in $\mathbf{C}$. Note that (2) is a generalization of (1) and reduces to it for the case of one-hot coding. If one sets $\sigma = \psi$ in (2) and uses $\mathbf{C} = \mathbf{I}_M$, then it is easily seen that $p_\sigma(k) = p_\psi(k)$ for all $k$. Figure 1(b) illustrates $p_\sigma$. The codeword assignment to classes, axes and colors in this figure are identical in meaning to those for Figure 1(a). Two crucial points emerge from this figure. One, in contrast to $p_\psi$, $p_\sigma$ allocates non-trivial volume in logit space to uncertainty, i.e. where $p_\sigma(0) \approx 0.5$. Two, $p_\sigma$ effectively shrinks the attack surface available to an attacker seeking to craft adversarial examples. Figure 1(c) illustrates this. Suppose the input $\mathbf{x}$ to the network has corresponding logits given by the magenta circle. $\mathbf{x}$ is such that $p_\psi(0|\mathbf{x}) \approx p_\sigma(0|\mathbf{x}) \approx 1$. Now consider 3 different adversarial perturbations of $\mathbf{x}$ to some $\mathbf{x}'$, whose corresponding logit perturbations are shown by the 3 arrows in the figure. For the perturbations given by the black arrows, $p_\psi(1|\mathbf{x}') \approx 1$, i.e., the class label under softmax is confidently flipped; but $p_\sigma(1|\mathbf{x}') \approx 0.5$, i.e., under $\sigma$ the model is now uncertain. Only the perturbation indicated by the gray (diagonal) arrow leads to $p_\sigma(1|\mathbf{x}') \approx 1$ (as well as $p_\psi(1|\mathbf{x}') \approx 1$). Fewer perturbation directions in logit space can (confidently) fool the classifier; the adversary must now search for perturbations to $\mathbf{x}$ which simultaneously decrease $z_0$ while increasing $z_1$.

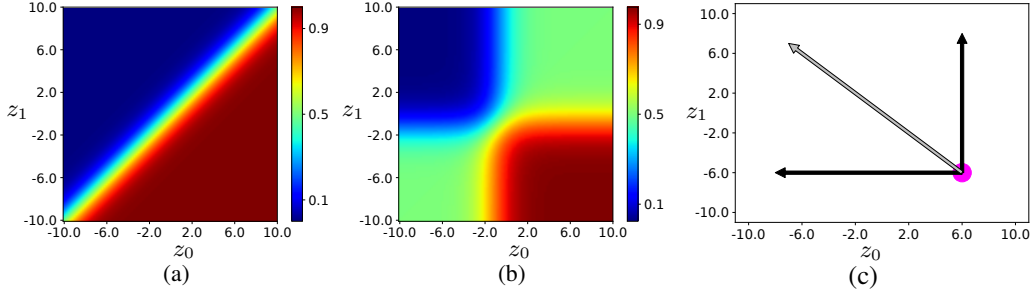

Figure 1: Probability of class 0 as a function of logits, for the (a) softmax activation and (b) sigmoid decoding scheme. (c). Movements in the space of logits from the original point (magenta circle) to new points (given by arrows); only the perturbation given by the gray arrow confidently fools the sigmoid decoder, while all perturbations confidently fool the softmax decoder.

## 2.3 Hamming distance

The Hamming distance between any two binary codewords $\mathbf{x}$ and $\mathbf{y}$, denoted $d(\mathbf{x}, \mathbf{y})$, is simply $|\mathbf{x} - \mathbf{y}|_0$, where $| \cdot |_0$ denotes the $L_0$ norm. The Hamming distance of codebook $\mathbf{C}$ is defined as

$$d = \min\{d(\mathbf{x}, \mathbf{y}) : \mathbf{x}, \mathbf{y} \in \mathbf{C}, \mathbf{x} \neq \mathbf{y}\} \tag{3}$$

The standard one-hot coding scheme has a Hamming distance of only 2. Practically, this means that if the adversary can sufficiently alter even a single logit, an error may occur. In Figure 1(b), for example, changing a single logit (i.e. an axis-aligned perturbation) is sufficient to make the classifier uncertain. Ideally, we want the classifier to be robust to changes to *multiple* logits.

What happens if we increase the Hamming distance between codewords? Consider the $M = N = 32$ case where each of 32 classes is represented by a 32-bit codeword (meaning that the DNN has 32 outputs versus 2 for the case in Figure 1). Figure 2 shows the probability of class 0 as a function of a 3 dimensional slice of the logits $(z_{29}, z_{30}, z_{31})$, where the other logits $z_i$ are fixed to $3\gamma(\mathbf{C}(0, i))$ where $\mathbf{C}(0, i)$ denotes the $i$th element of codeword 0 and $\gamma(x)$ is defined as 1 if $x > 0$ and $-1$ otherwise. (In other words, the fixed logits are set to be consistent with class 0). The colors in this figure are identical in meaning to those in Figure 1. For reference, the magenta circle shown has probability $> 0.999$ of being in class 0. The left-most column shows the probability of class 0 under the softmax activation and code $\mathbf{C} = \mathbf{I}_{32}$. The middle column uses the sigmoid decoding scheme with logistic activation and code $\mathbf{C} = \mathbf{I}_{32}$. The right-most column uses the sigmoid decoding scheme with tanh activation and code $\mathbf{C} = \mathbf{H}_{32}$, a Hadamard code of length 32. Within each column, two different views of the same logit space are shown. Note that for the softmax decoder, local perturbations within the logit space exist, (e.g., moving in an axis-aligned direction from the magenta point) which reduce the probability of class 0 to near 0. Also note how the softmax decoder has a very small region corresponding to uncertainty (i.e., probability near 0.5); as the logits vary, the model rapidly transitions from assigning probability $\approx 1$ to class 0 to assigning probability $\approx 0$. In contrast, the logistic decoder assigns far more volume to uncertainty. The Hadamard code based decoder is even more robust; it still assigns large probability to class 0 despite large changes to multiple logits.

Figures 1 and 2 illustrate the fact that with the softmax, a "small" change in logits $\delta \mathbf{z}$ can lead the model from being very certain of one class to being very uncertain of that class (and indeed, certain of another); sigmoid decoding with Hadamard codes greatly alleviates this problem. How does this relate to small changes in the input, $\delta \mathbf{x}$? Let $\mathbf{J}$ denote the Jacobian matrix of logits $\mathbf{z}$ with respect to input $\mathbf{x}$. By Taylor's theorem we know that $\delta \mathbf{z} \approx \mathbf{J} \cdot \delta \mathbf{x}$ and so $||\delta \mathbf{z}|| \leq ||\mathbf{J}|| \cdot ||\delta \mathbf{x}||$ where $|| \cdot ||$ denotes Euclidean norm for a vector and operator norm for a matrix. Assume that $||\mathbf{J}||$ is comparable across softmax and sigmoid schemes and choices of $\mathbf{C}$ (a fact we have empirically observed across several datasets). Then, in order to gain robustness to perturbations $\delta \mathbf{x}$, we can try to reduce $||\mathbf{J}||$; indeed this is the effect of most existing adversarial defenses. With our approach, in contrast, a larger $\delta \mathbf{z}$ is needed to move in logit-space from one class to another; hence a larger $\delta \mathbf{x}$ is needed.

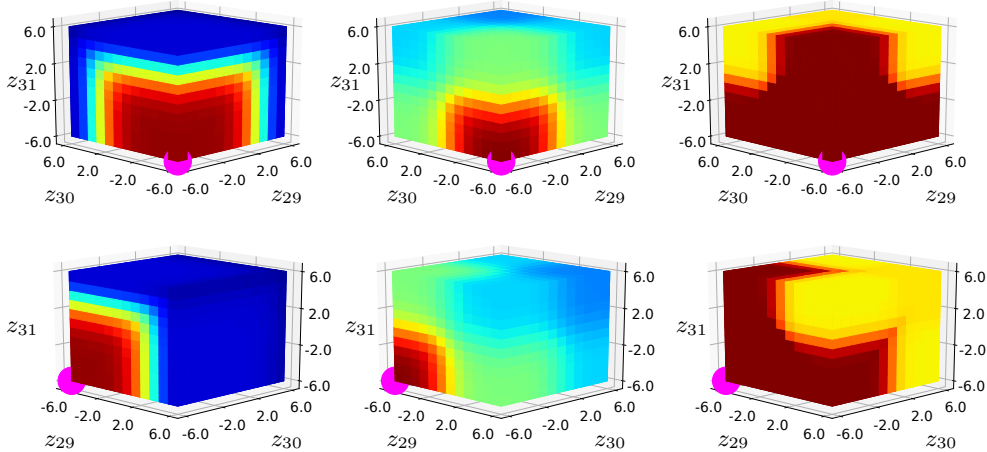

Figure 2: Probability of class 0 as a function of logits, for different choices of output activation and code, for a 32 class multi-classification problem: (leftmost column) softmax (Eq 1) with $\mathbf{C} = I_{32}$, (middle column) sigmoid decoder (Eq 2) with logistic activation and $\mathbf{C} = I_{32}$, (rightmost column) sigmoid decoder (Eq 2) with tanh activation and $\mathbf{C} = H_{32}$, a Hadamard code. 29 logit values are fixed and remaining logits (here denoted $z_{29}, z_{30}, z_{31}$) are allowed to vary. Colorbar is same as in Figure 1. Different choices of output activation and output code result in fundamentally different mappings of Euclidean logit space to class probabilities. Further details are in the main text.

## 2.4 Code design

We now turn to the choice of $\mathbf{C}$, which has been studied under the name of *error correcting output codes* (ECOC), popularized in the machine learning literature by [19]. The work therein and much of the work that has followed on ECOC focused on the potential gains in generalization in multi-class settings over conventional one-hot (equivalently, one-vs-rest) coding. Much of this research used ECOCs with decision trees or shallow neural networks. With the advent of deep learning and vastly improved accuracies even with conventional one-hot encodings, ECOCs are not in mainstream use. Several methods exist in order to create "good" ECOC which focus primarily on achieving a large Hamming distance between codes. A library implementing various heuristics, some inspired from coding theory, is available in [20]. Ideally, we would use $\mathbf{C}$ with the largest possible $d$ from (3) (though other factors, like good column separation, are also important). We first state a theorem which we use to select a near optimal choice for $\mathbf{C}$.

**Theorem 1** (*Plotkin's Bound*). *For an $M \times N$ coding matrix $\mathbf{C}$, $d \leq \left\lfloor \frac{N}{2} \frac{M}{M-1} \right\rfloor$*

Theorem 1 upper bounds the Hamming distance of $\mathbf{C}$. For $M$ large and $N$ even, the bound approaches $\frac{N}{2}$ which can be achieved if we choose $\mathbf{C}$ to be a Hadamard matrix. This choice has an important fortunate benefit. Recall that we would like to obtain probability estimates from our output, not just a classification decision. We say that our probability estimation is *admissible* if, whenever the network outputs any given codeword exactly, say $\mathbf{C}(j)$, the probability as computed by (2) is $p_\sigma(j) = 1$. If $\mathbf{C}$ is non-orthogonal, then $\mathbf{C}(j)$ may have positive correlation with $\mathbf{C}(i)$, in which case $p_\sigma(j) < 1$ even if the network outputs $\mathbf{C}(j)$. Thus, orthogonal $\mathbf{C}$ is required for admissible probability estimates. In this paper, we will use the notation $\mathbf{H}_P$ to denote a $P \times P$ Hadamard matrix. When there are more codewords $P$ available than actual classes $M$ (e.g., $P = 16$, $M = 10$ for CIFAR10), we simply select the first $M$ rows of $\mathbf{H}_P$ as codewords. More sophisticated optimizations are possible which also examine, for example, the correlation structure of the columns; we leave this for future work.

## 2.5 Bit independence

In a typical DNN, a single network outputs all the bits of the output code in its final layer (e.g., for MNIST, the final layer would be comprised of 10 neurons). However, it is also possible to

Table 1: Table characterizing various models tested in this paper

| Model | Architecture | Code | Probability estimation | $\sigma$ |
|---|---|---|---|---|
| Softmax | Standard | $\mathbf{I}_{10}$ | eq (1) | softmax |
| Logistic | Standard | $\mathbf{I}_{10}$ | eq (2) | logistic |
| Tanh16 | Standard | $\mathbf{H}_{16}$ | eq (2) | tanh |
| LogisticEns10 | Ensemble | $\mathbf{I}_{10}$ | eq (2) | logistic |
| TanhEns16 | Ensemble | $\mathbf{H}_{16}$ | eq (2) | tanh |
| TanhEns32 | Ensemble | $\mathbf{H}_{32}$ | eq (2) | tanh |
| TanhEns64 | Ensemble | $\mathbf{H}_{64}$ | eq (2) | tanh |
| Madry | Standard | $\mathbf{I}_{10}$ | eq (1) | softmax |

learn an ensemble of networks, each of which outputs a few bits of the output code. The errors in the individual bits that are made by a DNN or an ensemble method are often correlated; an input causing an error in one particular output often correlates to errors in other outputs. Such correlations reduce the effective Hamming distance between codewords since the dependent error process means that multiple bit flips are likely to co-occur. Therefore, promoting diversity across the constituent learners is crucial and is generally a priority in ensemble-based methods; various heuristics have been proposed, including training each classifier on a subset of the input features [21] or rotating the feature space [22]. The problem of correlation across ensemble members is more serious when each member solves the *same* classification problem; however, for ECOCs, each ensemble member $j$ solves a *different* classification problem (specified by the $j$th column of $\mathbf{C}$). Thus we find that it is sufficient to simply train an ensemble of networks, where each member outputs $B \ll N$ bits (neurons) of the output code; a diagram of the architecture used for experiments in this paper is given in Figures S1 and S2 in the supplement. In this paper, for codes whose length is a multiple of $4$ (all Hadamard codes), we set $B = \frac{N}{4}$. Else, we set $B = \frac{N}{2}$. Since each ensemble member shares no parameters with any other, the resulting architecture has reduced error correlations compared to a typical fully connected output layer.

## 3 Experiments

Our approach is general and dataset-agnostic; here we apply it to the MNIST and CIFAR10 datasets. All of our code is available at [23]. MNIST is still widely studied in adversarial machine learning research since an adversarially robust solution remains elusive. We conduct experiments with a series of models which vary the choice of code $\mathbf{C}$, the length of the codes $N$, and the activation function applied to the logits. Our training and adversarial attack procedures are standard; details are given in the supplement. Table 1 summarizes the various models used in this paper. "Standard" refers to a standard convolutional architecture with a dense fully connected output layer illustrated in the supplement in Figure S1, while "ensemble" refers to the setup described in Section 2.5 and illustrated in Figure S2. The final column describes the sigmoid function used in Eq (2). The "Madry" model is the adversarially trained model in [8]. Table 2 shows the results of our experiments on MNIST. The first column contains a descriptive name for the model (which is detailed in Table 1). Column 2 shows the total number of parameters in the model. Column 3 reports accuracy on the test set. The remaining columns show results on various attacks; all such results are in the white-box setting (adversary has full access to the entire model). Columns 4 and 5 show results for the projected gradient descent (PGD, $\epsilon = 0.3$) and Carlini-Wagner (CW) attacks [24], respectively, These columns show the fraction of adversarially crafted inputs which the model correctly classifies, i.e., examples which fail to be truly adversarial. Column 6 contains results of the "blind spot attack" [25], which first scales images by a constant $\alpha$ close to 1 before applying the Carlini Wagner attack. Column 7 shows results for the 'Distributionally Adversarial Attack" (DAA) [26] (which is based on the Madry Challenge leaderboard [27]). We choose this attack since it appeared (as of mid 2019) near or atop the leaderboards for both MNIST and CIFAR10 datasets. Column 8 shows the fraction of random inputs for which the model's maximum class probability is smaller than $0.9$; here, a random input is one where each pixel is independently and uniformly chosen in $(0, 1)$. Column 9 shows the accuracy on test inputs where each pixel is independently corrupted by additive uniform noise in $[-\gamma, \gamma]$, where $\gamma = 1$ $(0.1)$ for MNIST (CIFAR10) and clipped to lie within the valid input pixel range, e.g. $(0, 1)$.

Several points of interest emerge from the results in Table 2. One, the Logistic model is superior to the Softmax model due to the phenomena illustrated in Figure 1(b) and (c); in particular, the result on Random attacks indicates the Logistic indeed goes a long way towards reducing the irrational overconfidence of the softmax activation. Two, Tanh16's superior performance over Logistic shows the advantage of using a code with larger Hamming distance. Three, LogisticEns10's vastly improved performance on Random attacks shows the importance of reduced correlation among the output bits (described in Section 2.5). Four, TanhEns16 shows a marked improvement across all dimensions over all predecessors; it combines the larger Hamming distance with reduced bit correlation. TanhEns32 shows results for a 32 output codes; we find that performance appears to plateau and that increased code length confers no meaningful additional benefit for this dataset. In general, we might expect diminishing gains in performance with increasing code length relative to number of clases. Finally, comparing all the ensemble (ending in "Ens") models to the Madry model, we see the latter uses many more parameters. The TanhEns16 model has superior performance to Madry's model on all attacks, sometimes significantly so. Also note that while Madry model's benign accuracy is much lower than the state-of-the-art for MNIST, the TanhEns16 model enjoys excellent accuracy.

Figure 3(a)-(c) compares the probability distributions of various models on MNIST for (a) benign, (b) projected gradient descent (PGD) generated adversarial, and (c) random examples. In more detail, for each example $\mathbf{x}$, we compute the probability that the model assigns to the most probable class label of $\mathbf{x}$. We compute and plot the distribution of these probabilities over a randomly chosen set of 2000 test examples of MNIST. Figure 3(a) shows that all models assign high probability to nearly all (benign) inputs, which is desirable since all models have high test set accuracy. Figure 3(b) compares models on adversarial examples. The TanhEns16 and TanhEns32 models tend to (correctly) be less certain than the other models (note that these models have bimodal distributions; the lower (upper) mode tends to correspond to adversarial examples that do (not) resemble the nominal class given by the model). Figure 3(c) compares models on randomly generated inputs. While the Softmax and Madry models are often certain of their decisions, the other models, particularly the TanhEns16 and TanhEns32, correctly put most mass on low probabilities. In summary, Figure 3 shows that the TanhEns model family has two highly desirable properties: 1) like the Softmax and Madry models, it is very certain about the (correct) label on benign examples, and 2) unlike the Softmax and Madry models, it is often uncertain about the (incorrect) label on adversarial and random examples. Furthermore, when TanhEns is certain (uncertain), the example often resembles the target class (no recognizable class); see Figures S2 and S3 in the supplement for sample illustrations. Taken together, these facts suggest that the TanhEns model class yields very good probability estimates.

Table 3 is analogous to Table 2, but presents results for CIFAR10. Figure 3(d)-(f) shows the probability distributions for CIFAR10. For CIFAR10, our baseline is Madry's adversarially trained CIFAR10 model. We notice results that are all qualitatively similar to those in the MNIST case; again, the TanhEns model family has strong performance and is competitive with or outperforms Madry's model. A key distinction is that now, 32 and 64 bit codes show clear improvements over 16 bit codes. Further improvements to the TanhEns performance are likely possible by using more modern network architectures; we leave this for future work.

Finally, Figure 4 plots model accuracy versus the PGD $L_\infty$ perturbation limit $\epsilon$, for both (a) MNIST and (b) CIFAR10. The TanhEns models dominate Madry's model. Notably for MNIST, the accuracy drops significantly around $\epsilon = 0.5$; this is to be expected since at this value of $\epsilon$, a perturbation which simply sets all pixel values to $0.5$ (therby creating a uniformly grayscale image) will obscure the true class. Because model accuracy rapidly drops to near 0 as $\epsilon$ grows, the figure provides crucial evidence that our approach has genuine robustness to adversarial attack and is not relying on "gradient-masking" [28]. Also, the TanhEns models significantly outperform Madry's model for $\epsilon > 0.3$ ($\epsilon > 0.031$) on MNIST (CIFAR10), indicating that our model has an intrinsic and wide-ranging robustness which is not predicated on adversarially training at a specific level of $\epsilon$.

## 4   Conclusion

We have presented a simple approach to improving model robustness that is centered around three core ideas. One, moving from softmax to sigmoid decoding means that a non-trivial volume of the Euclidean logit space is now allocated towards model uncertainty. In crafting convincing adversarial perturbations, the adversary must now guard against landing in such regions, i.e. his attack surface is smaller. Two, in changing the set of codewords from $\mathbf{I}_M$ to one with larger Hamming distance, the

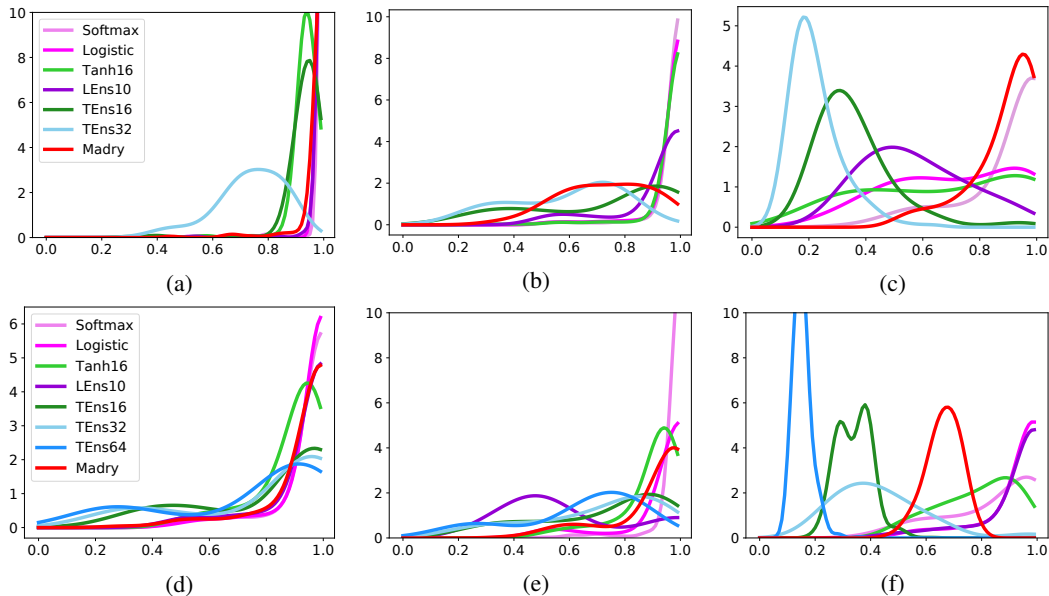

Figure 3: Distribution of probabilities assigned to the most probable class on the test set of (a-c) MNIST and (d-f) CIFAR10, by various models. LogisticEns10 and TanhEns models are abbreviated as LEns10 and TEns, respectively. $x$ axis is the probability assigned by the classifier, $y$ axis is the probability density. Legend in first column is common to all figures. (a) and (d). Distribution of probabilities on benign (non-adversarial) examples. (b) and (e). Distribution of probabilities on adversarial examples. (c) and (f). Distribution of probabilities on randomly generated examples where each pixel is sampled independently and uniformly in $[0, 1]$.

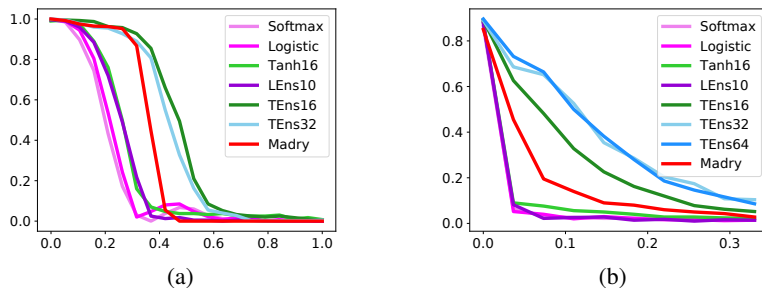

Figure 4: Model accuracy ($y$-axis) versus perturbation strength $\epsilon$ ($x$-axis) for (a) MNIST and (b) CIFAR10. LogisticEns10 and TanhEns models are abbreviated as LEns10 and TEns, respectively. Curves are based on attacking a random sample of 200 test samples.

Table 2: Accuracies of various models trained on MNIST against various attacks. "-" indicates experiment was not performed.

| Model | # Params | Benign | PGD | CW | BSA $\alpha = 0.8$ | DAA | Rand | +U(-1,1) |
|---|---|---|---|---|---|---|---|---|
| Softmax | 330, 570 | .9918 | .082 | .540 | .180 | - | .270 | .785 |
| Logistic | 330, 570 | .9933 | .093 | .660 | .210 | - | .684 | .829 |
| Tanh16 | 330, 960 | .9931 | .421 | .790 | .320 | - | .673 | .798 |
| LogisticEns10 | 205, 130 | .9933 | .382 | .880 | .480 | - | .905 | .812 |
| TanhEns16 | 401, 168 | .9948 | .929 | 1.0 | 1.0 | .923 | .988 | .827 |
| TanhEns32 | 437, 536 | .9951 | .898 | 1.0 | 1.0 | - | 1.0 | .858 |
| Madry | 3, 274, 634 | .9853 | .925 | .840 | .520 | .888 | .351 | .150 |

Table 3: Accuracies of various models trained on CIFAR10 against various attacks. "-" indicates experiment was not performed.

| Model | # Params | Benign | PGD | CW | BSA $\alpha = 0.8$ | DAA | Rand | +U(-.1,.1) |
|---|---|---|---|---|---|---|---|---|
| Softmax | $775,818$ | .864 | .070 | .080 | .040 | - | .404 | .815 |
| Logistic | $775,818$ | .865 | .060 | .140 | .100 | - | .492 | .839 |
| Tanh16 | $776,208$ | .866 | .099 | .080 | .100 | - | .700 | .832 |
| LogisticEns10 | $1,197,978$ | .877 | .100 | .240 | .140 | - | .495 | .852 |
| TanhEns16 | $2,317,456$ | .888 | .515 | .760 | .760 | .514 | .999 | .842 |
| TanhEns32 | $2,631,456$ | .891 | .574 | .780 | .770 | .539 | .989 | .869 |
| TanhEns64 | $3,259,456$ | .896 | .601 | .760 | .760 | .543 | 1.0 | .875 |
| Madry | $45,901,914$ | .871 | .470 | .080 | 0.0 | .447 | .981 | .856 |

Euclidean distance in logit space between any two regions of high probability for any given class becomes larger. This means that the adversary's perturbations now need to be larger in magnitude to attain the same level of confidence. Three, in learning output bits with multiple disjoint networks, we reduce correlations between outputs. Such correlations are implicitly capitalized on by common attack algorithms. This is because many attacks search for a perturbation by following the loss gradient, and the loss will commonly increase most rapidly in directions where the perturbation impacts multiple (correlated) logits simultaneously. Importantly, since it simply alters the output encoding but otherwise uses completely standard architectural components (i.e., convolutional and densely connected layers), the primary source of our approach's robustness does not appear to be obfuscated gradients [15].

The learner that results is surprisingly robust to a variety of non-benign inputs. Our approach has many interesting and complementary advantages to existing approaches to adversarial defense. It is extremely simple and integrates seamlessly with existing machine learning pipelines. It is extremely fast to train (e.g., it does not rely on in-the-loop adversarial example generation) and during inference time (compared to, e.g., manifold based methods or generative models which often involve a potentially costly step of computing the probability of the input under some underlying model).In the models for MNIST and CIFAR10 studied in this paper, our networks use far fewer parameters than the Madry model. Because our model is not adversarially trained with respect to any $L_p$ norm attack, it appears to have strong performance across a variety of adversarial and random attacks. This bodes well for our approach to generalize to future attacks. Another significant advantage is that our approach has no apparent loss on benign test set accuracy, in major contrast to other adversarial defenses. Finally, further gains are achievable by increasing the diversity across ensemble members, such as training each ensemble member on different rotations [22] or with distinct architectures.

Our model also yields vastly improved probability estimates on adversarial and garbage examples, tending to give them low probabilities; this is particularly interesting since attempts at using Bayesian neural networks to improve probability estimation on adversarial examples have not found clear success yet [29]. It is well known that using the standard softmax to convert logits to probabilities leads to poor estimates [30]; approaches such as Platt scaling which improve probability calibration on the training manifold still produce overconfident estimates on adversarial and noisy inputs. While we have not carefully studied our model's probability calibration, we have presented strong empirical evidence suggesting much improved estimates should be achievable both on and off the training manifold.

One important avenue for further study is to consider datasets of larger input dimensionality, such as ImageNet. It may be possible that in very high input dimensions, adversarial perturbations exist that can still surmount the larger Hamming distances afforded by ECOCs (though our results here provide hope that the labels of any such examples will typically have lower probability). However, a counter to this might simply involve using longer codes; our experiments with CIFAR10 indicate this could be a viable strategy. Such an approach would tradeoff training time for robustness, reminiscent of the tradeoff in communications theory between data rate and tolerance to channel errors. A second avenue for further research is to combine our idea with existing methods based on adversarial training or with provable approaches to certified robustness [11]. We believe that our approach will make any other adversarial defense much stronger.

## Footnotes

[1]Strictly speaking, the noise in coding theory is stochastic in nature, while adversarial perturbations are non-random. Nonetheless, our analysis and results indicate there is significant benefit in taking this view.

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
