[Supplementary Material · VermaSwami_NeurIPS2019_Supplement.pdf]

# Supplement: Error Correcting Output Codes Improve Probability Estimation and Adversarial Robustness of Deep Neural Networks

**Gunjan Verma**
CCDC Army Research Laboratory
Adelphi, MD 20783
gunjan.verma.civ@mail.mil

**Ananthram Swami**
CCDC Army Research Laboratory
Adelphi, MD 20783
ananthram.swami.civ@mail.mil

## 1 Experiments

### 1.1 Training

All models (e.g, all the models shown in Table 1 of the main text, except for Madry's model) are trained for $E$ epochs using the Adam optimizer and a learning rate of $3\mathrm{e}{-}4$, where $E = 150$ for MNIST and $E = 400$ for CIFAR10. To prevent overfitting, we employ two simple procedures. One, we add zero-mean Gaussian noise with standard deviation 0.3 for MNIST and 0.032 for CIFAR10, respectively, to every pixel. Also, instead of training on the original images, we train instead on images of the form $(1 - \alpha)\mathbf{x} + \alpha\mathbf{y}$ where $\mathbf{x}$ is the original image and $\mathbf{y}$ is a randomly chosen image from the training set. We set $\alpha = 0.3$ for MNIST and $\alpha = 0.032$ for CIFAR10. These values of $\alpha$ are chosen simply to be equal to the corresponding parameter $\epsilon$ used in the $L_\infty$ PGD attack. This latter procedure intuitvely appears to be a useful form of data augmentation for adversarial machine learning contexts (though we find it provides minimal benefit against adversarial attack; the low accuracies against PGD attack for the Softmax and Logistic models results on MNIST demonstrate this). For CIFAR10, we also use standard data augmentation (e.g., rotating and shifting images). For Softmax models, we use the standard cross-entropy loss. For Logistic models, we use the binary cross-entropy loss (equivalently, the logistic loss) on a per output neuron (bit) basis. For Tanh models, which operate with Hadamard codes containing $+/-1$ elements (instead of the usual 1 and 0), we use the (SVM) hinge loss on a per output neuron basis.

### 1.2 Adversarial attack

In order to generate adversarial attacks on our models, we use the CleverHans 3.0.1 software [1]. In all cases, we use the default values of attack parameters with changes noted herein; when we do change attack parameters, we always do so as to make the attack strictly stronger. We use the projected gradient descent (PGD) attack [2] for $L_\infty$ based attacks, which has a parameterized maximum pixel-wise distortion which we denote by $\epsilon$; in this paper we use $\epsilon = 0.3$ for MNIST and $\epsilon = 0.031$ $(\frac{8}{255})$ for CIFAR10. We use 500 iterations per PGD attack for MNIST and 200 iterations for CIFAR10. For $L_2$ attacks we use the Carlini-Wagner (CW) algorithm [3] with a learning rate of $1\mathrm{e}{-}3$ and 10 binary search steps. Benign accuracy is calculated on the entire test set. For PGD attacks, we use 2000 test set examples to compute adversarial accuracy. For CW attacks, we use 100 test set examples; we use fewer examples for CW as it is a significantly slower attack to execute. "Random" attacks involve simply generating inputs with pixels independently and uniformly sampled from $[0, 1]$. The blind-spot attack (BSA) [4] of Table 2 in the main text simply applies the CW attack to a scaled version of the input $\alpha\mathbf{x}$ where $\alpha \approx 1$, $\alpha < 1$.

Figure S1: Architecture of "Standard" models. The processing flow is from left to right. The notation inside each box indicates the layer description; e.g. the left-most box indicates that input is convolved with $F_1$ filters of shape $3 \times 3$ with stride $1$. The yellow dashed rectangle indicates that the components within it are repeated $R_1$ times in succession in the architecture. The solid blue color indicates the component is repeated $R_2$ times in succession. After the first 2 convolutional components, the network is followed by 3 dense (i.e., fully connected) layers. The final dense layer consists of $D_3$ neurons with linear activation. For MNIST, we use $R_1 = 3$, $R_2 = 1$, $F_1 = 64$, $D_1 = 128$, $D_2 = 64$, $D_3 = N$ where $N$ is the output codeword size. For CIFAR10, we use $R_1 = 3$, $R_2 = 2$, $F_1 = 64$, $D_1 = 128$, $D_2 = 64$, $D_3 = N$ where $N$ is the output codeword size.

Figure S2: Architecture of "Ensemble" models. The processing flow is from left to right. The notation inside each box indicates the layer description; e.g. the left-most box indicates that input is convolved with $F_1$ filters of shape $3 \times 3$ with stride $1$. The yellow dashed rectangle indicates that the components within it are repeated $R_1$ times in succession in the architecture. The solid blue color indicates the component is repeated $R_2$ times in succession. After the first 2 convolutional components, the network splits into four branches. Each branch applies convolution and densely connected layers. Finally, a neuron with linear activation outputs the logit (shown as a circle). To output $N$ bits, this entire unit is replicated $K$ times and each is separately trained. The final layer contains 4 neurons in this figure for illustration purposes; in general it will contain $\frac{N}{4}$ bits if $N$ is a multiple of 4 and we set $K = 4$; else, the final layer is comprised of $\frac{N}{2}$ bits and we set $K = 2$. The intuition behind the branching is to help de-correlate the errors made by the output bits, since no parameters are shared across branches. For MNIST, we use $R_1 = 3$, $R_2 = 1$, $F_1 = 32$, $F_2 = 4$, $D_1 = 16$, $D_2 = 8$. For CIFAR10, we use $R_1 = 3$, $R_2 = 2$, $F_1 = 64$, $F_2 = 16$, $D_1 = 16$, $D_2 = 8$.

### 1.3 Standard architecture

Figure S1 shows the architecture for "Standard" models (in the nomenclature of Table 1 of the main text). This is essentially a vanilla convolutional neural network architecture.

### 1.4 Ensemble model architecture

Figure S2 illustrates the basic architecture used for experiments with "ensemble" models (those ending in "Ens" in Table 2 of the main text). The caption therein contains the details. This neural network outputs the logits corresponding to $4$ "bits" of the output code. More generally, for a codeword with $N$ bits which is a multiple of $4$ (all Hadamard codes), an ensemble of $4$ unique instantiations of the network in Figure S2 are learned; else 2 unique instantiations are learned. The reason for learning in this way (as opposed to learning all $N$ output neurons in a single network, as is conventionally done) is to reduce dependence across output bits as discussed in the main text. In particular, we find empirically that this setup leads to better performance on "Random" attacks as well as reduces the probability that the model is (incorrectly) certain on adversarial examples. It is of course also possible to learn in other variations, for example to learn one bit per network (for a total of $N$ networks). Each of these designs has a tradeoff between how "dependent" bits are and the number of parameters used. Note that it may seem that since we are learning multiple networks, we will require a very large number of parameters. However, this is not necessarily the case; this is because each constituent network solves a smaller classification problem than a conventional DNN. Therefore, the constituent networks used in our method can use fewer parameters per network.

### 1.5 Figures

Here we show some randomly selected samples of adversarial examples on the TanhEns16 model. We attack with PGD, $\epsilon = 0.3$ the TanhEns16 model and sample from adversarial examples with probability $> 0.9$ (Figure S3), or with probability $< 0.25$ (Figure S4). The key point to take away from these figures is that the decisions (classifications as well as probabilities assigned) made by the model are "reasonable" and ones a human would likely agree with. While assessing the quality of probabilities produce by a model is largely a subjective exercise, these figures, along with Figure 3 in the main text lend support to the notion that the TanhEns16 produces good probability estimates.

### 1.6 Time Complexity

Since our method only involves changing the output encoding, our training times are comparable to those for training a conventional DNN of the same architecture. Our method adds no overhead. Instead of learning a single network predicting $N$ bits, we learn 4 networks each predicting $\frac{N}{4}$ bits. This generally increases training time by a constant factor if no parallelization is used. However, since there are no interdependencies between constituent networks, trivial parallelization across cores or GPUs is achievable and can result in effectively faster training. If longer codes are desired, one can simply train additional networks and augment them to existing learned models.

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

Figure S3: High confidence adversarial examples. We plot one randomly selected adversarial example from each true class label (0 - 9) under PGD($\epsilon = 0.3$) attack of 2000 test examples that are misclassified with high confidence (probability > 0.9) by the TanhEns16 model. No high-confidence adversarial attacks on classes $1, 5$ were found in the attack. (a) True: 0. Predicted: 7. (b) True: 2. Predicted: 7.(c) True: 3. Predicted: 5. (d) True: 4. Predicted: 9. (e) True: 6. Predicted: 0. (f). True: 7. Predicted: 9. (g). True: 8. Predicted: 4. (h). True: 9. Predicted: 4. The example in (h) is likely mislabeled in the test set. In all cases, it seems plausible (from a human perceptual viewpoint) why the model might be confident about the predicted class.

Figure S4: Low confidence adversarial examples. We plot one randomly selected adversarial example from each true class label (0 - 9) under PGD($\epsilon = 0.3$) attack of 2000 test examples that are misclassified with low confidence (probability < 0.25) by the Tanh(16) model. No low-confidence example of class 2 was found in the attack. (a) True: 0. Predicted: 7.(b) True: 1. Predicted: 6. (c) True: 3. Predicted: 1. (d) True: 4. Predicted: 6. (e). True: 5. Predicted: 3. (f). True: 6. Predicted: 8. (g). True: 7. Predicted: 2. (h). True: 8. Predicted: 5. (i). True: 9. Predicted: 8. In all cases, the numbers are confusing for a human too, so the low confidences are "understandable".