[Reviews · NeurIPS 2019]

Reviewer 1



Originality: While the methods used in the paper exist already, I think the way in which they have been combined is sufficiently novel and interesting. Quality: I think the idea is theoretically sound and is evaluated sufficiently. I agree with the intuition that encoding logits according to a Hadamard matrix provides sufficient separation between logits, which presumably makes it harder for an adversary to misclassify and input. I assume that when the authors say whitebaacks, they reveal the Hadamard matrix and final layer to the adversary. The idea of using an ensemble of networks where each network predicts two bits is well motivated. Clarity: I think the paper is very well written and could possibly be reproduced by others. Significance: For me the most significant part of the paper are Tables 1, 2 and 3. I think the reported robustness and improvements over Madry is a convincing result. Clarifications/weaknesses: 1. When the authors say whitebaacks, I assume this means that the adversary can see the full network with the final layers, every network in the ensemble, every rotation used by networks in the ensemble. I would like them to confirm this is correct. 2. Did the authors study numbers of bits in logits helps against a larger epsilon in the PGD attack? Because intuition suggests that having a 32 bit logit should improve robustness against a more powerful adversary. This experiment isn't absolutely necessary, but does strengthen the paper. 3. Did the authors study the same approach on Cifar? It seems like this approach should be readily applicable there as well. ---Edit after rebuttal--- I am updating my score to 8. The improved experiments on Cifar10 make a convincing argument for your method.

Reviewer 2



Coding theory addresses both stochastic and adverarial noise. Minimum distance, one of the simplest and most familiar metrics of a code, measure the ability of a code to correct adversarial errors. I don't understand the argument in Section 2.1 about the volume of the "uncertain region" in the space of logits. Why is the Euclidian volume relevant? Any continuous function with the same limiting behavior will have this behavior. The relevant way to measure the size of the uncertain region is using the distribution of the inputs to the softmax layer. If the variance of these inputs is too large then the outputs will be overconfident and if the variance is too small they will be underconfident. A persuasive argument that the shape of the softmax function is responsible for overconfident classification must be more nuanced than the one made here. Section 2.2 describes a "correlation-based decoder". This is a logistic activation layer followed by a linear layed defined by the code followed by a ReLU layer followed by normalization to a probability distribution. Why were these two choices for the nonlinear activations made? Doesn't the ReLU throw away information that could be useful for calibrated confidence levels? Why is the alpha hyperparameter introduced? It is not used in the experiments and the framework does not seem to depend on it. In Section 2.3, the code described is equivalent to a repetition code. This is a trivial code. Why should it offer any benefit? The Hamming distance between codewords is increased, but on the other hand, the operator norm of the final layer is larger than that of an identity layer. This would seem to cancel out the benefit of the code. In general, the problem of adversarial examples exists even with only two classes, but there are no interesting codes in this case. Theorem 1 is a variation on the well known Plotkin bound.

Reviewer 3



Summary: The region of uncertainty (prediction probability close to 0.5) for softmax of logits is extremely small near an M-1 dimensional hyperplane in the logits space. The reason is changing one of the logits for one of the classes affects the probability vectors in all dimensions. The authors show that, if each logit is first converted to an independent probability using 1/(1+exp(-x)) function and the probability vector correlated with each codeword of an error correcting in a soft way to decode, this method has a large volume of uncertainty. The volume of uncertainty is larger when the min hamming distance of the code is large. This because multiple logits must be changed at the same time to cause a wrong decoding. Authors demonstrate this with nice plots. Authors propose to use subset of rows of Hamming Matrix for two properties: They have almost the best minimum hamming distance (half the dimension of the code) and they are orthogonal (which is important for unconfused decoding). They derive an upper bound on distance for any code. I checked the proof of the upper bound - it seems correct. Then the authors demonstrate performance (using Hamming 16 code) against PGD and Carlini Wagner attacks. In the case of MNIST for PGD attacks, it loses 7% accuracy compared to Madry et al's adversarialy trained examples. In all other combinations it seems to better. In Table 2 additional results are provided for epsilon=0.4,0.45 where it is much better than the Madry et al's adversarially trained models. For Fashion-MNIST, results are more encouraging against an adversarially trained MNIST. Supplement has very interesting insights as to how the codes help in various ways (adversarial examples having high uncertainty amongst others and actual example digits which are difficult even for humans and hence already on the border). Significance and Originality: Use of error correcting output codes to improve robustness with empirical results is very novel (although explored in Machine learning in some other context before). The results seem very promising as well. Clarity: I like the presentation of the paper, explaining the intuition behind using codes with large distance. Quality: Submission seems technically sound. Weaknesses: a) This page from the "Madry Challenge" - https://github.com/MadryLab/mnist_challenge - lists state of the art attacks. The one reported in Table 1 for epsilon=3 seems to be the PGD with 100 steps. What happens with other state of the art attacks ? - At least the latest in the list could have been tried instead of just PGD. b) Authors could have tried on CIFAR-10 - again - the leaderboard roughly has 47% accuracy for the adversarially trained model. It would have been interesting to find if output codes could contribute independently or in combination with existing robustification methods. c) Just based on the code + Lipschitz constant + architecture information is it possible to offer a robustness certificate ? Does using codes make it easier to certify ?? ****After the rebuttal ****** I asked for more powerful attacks and also results for CIFAR-10. The authors did exactly that. Under the DAA attack (which is the top 2 in CIFAR-10, MNIST) for CIFAR-10 their method achieves 55% which is 10% (!) more than 44% on Madry's adversarially trained model. That really is a big improvement. The authors argument about higher volume of the uncertainty region seems convincing and intuitive and the fact that one has to disturb multiple logits instead of just one single one in the softmax case is very intuitive. I thank the authors for their work even during the rebuttal in clarifying my questions about experiments. I am increasing my score.

[Author Response · NeurIPS 2019]

Paper ID: 4658. 0. We present results for more powerful attacks from Madry Challenge leaderboard for MNIST and CIFAR10. We chose "Distributionally Adversarial Attack" (DAA) by Zheng as it appears atop the leaderboards for both MNIST and CIFAR10 datasets. Results for the original Tanh(16) model are shown in Table 1. Results are also shown for an "improved" version of Tanh(16) model which uses more convolutional filters per layer (32 instead of 25), and Gaussian noise with std. dev $0.25$ (instead of the $0.15$). These changes improve robustness of the model.

1. The term "white-box" means that the adversary knows *everything* about the model, i.e. full ensemble with all layers and every rotation used by all ensemble members.

2. We present results in Table 1 for the case of 32 bit codes. Intuition indeed suggests this code should be more powerful. In general, though, we find robustness asymptotes with increasing code length; this appears related to the "rank" of the coding matrix. Increasing code length relative to number of classes likely results in increasingly correlated logits. Due to bit dependence, the effective Hamming distance does not grow with code length.

3. We feel that the Reviewer's contention that "Any continuous function with the same limiting behavior will have this behavior." is not relevant; we constructed a method to estimate probabilities (eq 3) which does not have this behavior.

4. Euclidian volume is relevant for uncertainty; consider the following. Let $y$ ($\delta y$) denote the softmax's logits (change) corresponding to an input $x$. Let $J$ denote Jacobian matrix of logit layer evaluated at $x$. By Taylor's theorem, $\delta y \approx J \times \delta x$. In general we don't know singular vectors of $J$, and $\delta_x$ is controlled by the adversary so can point in any direction. $J$ is likely to be full row rank so $\delta y$ can point in an arbitrary direction. Without further assumptions on adversary, if Euclidean volume (in $y$-space) associated with uncertainty is vanishingly small (as with softmax), it is easy to find $\delta x$ whose induced $\delta y$ moves into a region of high-confidence; i.e., the adversary can reach a region of high confidence by perturbing $x$ to $x + \delta x$ *regardless of $x$'s location in input space*. Empirical evidence of this: see Fig 3(c) in paper; softmax is usually highly confident even for an input which is random noise.

5. The Reviewer's suggestion to consider distribution of activations to the softmax layer seems reasonable for data *known to lie on the training manifold*. However, adversarial (and 'Random" inputs) are (way) off this manifold.

6. Rationale for design choices used for the correlation decoder are simply to yield a valid probability estimate (non-negative, sum to 0). The logistc maps (unnormalized) $z_k$ to a similar range as the code elements in $C$; ReLU ensures probability estimates are non-negative. We don't aim to achieve carefully calibrated probability estimates; we agree with the Reviewer that such an undertaking may reveal other designs which are better suited for precise calibration. Our results in Figures 3(a)-(c) indicate our probability estimates are still far better than those of conventional models.

7. Thank you to Reviewer for pointing out Theorem 1 is Plotkin bound; we will drop proof and cite in revision.

8. That using softmax to convert logits to probabilities is not a good idea has been established in many papers; see e.g., "On Calibration of Modern Neural Networks" by Guo et Al.). Our empirical results (e.g., Fig 3 in te main text) also clearly show this. Some corrective action is needed (such as Platt scaling). However, our tests indicate that Platt scaling still produces overconfident estimates on adversarial and noise inputs (it appears to only calibrate on the training manifold). By contrast, our approach appears well-behaved even off the training manifold.

9. The Lipschitz constant of the network is *not* larger due to code-induced widening. The final 2 layers of Madry's MNIST model are 1. Fully connected layer of $1024$ units. 2. Softmax layer of $10$ units (softmax layer). The final 2 layers of our model are 1. Fully connected layer of 16 units (i.e. "code" layer); 2. Final layer of 10 units (computes probability using eq (3) of main text). Compared to typical architectures, ours does *not* induce widening of the network.

10. The fact that we used 2 classes in Section 2.3 is not essential; key point is that larger Hamming distance (=4) increases Euclidean distance between high-probability regions; Point 4. above elaborates on utility of Euclidean distance. Reviewer's point is still well-taken, and in the revision we will reword this section to consider $M > 2$ class example using a Hadamard matrix of code length $8$ (which has Hamming distance $4$). Fig. 2 would then correspond to fixing 5 of the logits and examining the remaining 3. Such a code would still have a Hamming distance of 4, and the key idea that multiple logits (instead of a single logit as with softmax) need to be altered to effect a class change still holds.

11. We were unclear about meaning of Reviewer's comment on "the relationship between adversarial constraints at network input and adversarial constraints before decoding layer". But, our objective in the paper was to explicltly consider an adversary that was *minimally* constrained at the input (i.e. could generate $L_\infty$, $L_2$, rotations, noise attacks).

Table 1: Accuracies against various attacks; "-": experiment was not run. "*": training terminated due to time constraints. model was still learning; result will improve with more training. MNIST: $\epsilon = 0.3$; CIFAR10: $\epsilon = .031$

| Model | Dataset | Code | Benign | PGD $\epsilon = 0.3(.031)$ | PGD $\epsilon = 0.4$ | DAA $\epsilon = 0.3(.031)$ |
|---|---|---|---|---|---|---|
| Tanh(16) | MNIST | $\mathbf{H}_{16}$ | .9911 | .853 | .49 | 0.848 |
| Tanh(32) | MNIST | $\mathbf{H}_{32}$ | .9901 | .847 | .472 | 0.821 |
| Tanh(16)_Improved | MNIST | $\mathbf{H}_{16}$ | .9925 | .901 | .541 | 0.888 |
| Tanh(16) | CIFAR10 | $\mathbf{H}_{16}$ | .848* | .578 | - | .551 |
| Madry | CIFAR10 | $\mathbf{I}_{10}$ | .873 | .470 | - | .447 |

[Meta-Review · NeurIPS 2019]

This paper proposes the use of error correcting codes as class representations to improve robustness for adversarial attacks. The main idea of error correcting output codes is well-known, but this is the paper that shows that such ideas can be used for adversarial robustness. The paper shows very promising results especially in the rebuttal for CIFAR10. The distance bound is equivalent to Plotkin as the reviewer pointed out so this should be fixed in the paper.